# Collaborative Robot Precision Task in Medical Microbiology Laboratory

**DOI:** 10.3390/s22082862

**Published:** 2022-04-08

**Authors:** Aljaz Baumkircher, Katja Seme, Marko Munih, Matjaž Mihelj

**Affiliations:** 1Laboratory of Robotics, Faculty of Electrical Engineering, University of Ljubljana, Tržaška Cesta 25, 1000 Ljubljana, Slovenia; marko.munih@fe.uni-lj.si (M.M.); matjaz.mihelj@fe.uni-lj.si (M.M.); 2Institute of Microbiology and Immunology, Faculty of Medicine, University of Ljubljana, Zaloška 4, 1000 Ljubljana, Slovenia; katja.seme@mf.uni-lj.si

**Keywords:** learning from demonstration, kinesthetic teaching, sub-millimetre accuracy, mass spectrometry, MALDI, colony picking

## Abstract

This study focuses on the feasibility of collaborative robot implementation in a medical microbiology laboratory by demonstrating fine tasks using kinesthetic teaching. Fine tasks require sub-millimetre positioning accuracy. Bacterial colony picking and identification was used as a case study. Colonies were picked from Petri dishes and identified using matrix-assisted laser desorption/ionization (MALDI) time-of-flight (TOF) mass spectrometry. We picked and identified 56 colonies (36 colonies of Gram-negative *Acinetobacter baumannii* and 20 colonies of Gram-positive *Staphylococcus epidermidis*). The overall identification error rate was around 11%, although it was significantly lower for Gram-positive bacteria (5%) than Gram-negative bacteria (13.9%). Based on the identification scores, it was concluded that the system works similarly well as a manual operator. It was determined that tasks were successfully demonstrated using kinesthetic teaching and generalized using dynamic movement primitives (DMP). Further improvement of the identification error rate is possible by choosing a different deposited sample treatment method (e.g., semi-extraction, wet deposition).

## 1. Introduction

Clinical microbiology laboratories play a critical role in preventing, diagnosing and treating infectious diseases caused by microorganisms (e.g., bacteria, fungi). In order to identify which bacteria cause an infection, microbiologists often cultivate clinical samples on solid media in Petri dishes, after which they pick and prepare bacterial colonies according to the chosen identification method. In the past two decades, mass spectrometry (MS) became the commonly used methodology for identifying bacteria and fungi, since it is accurate, faster, and less labour-intensive than other methods [1]. Matrix-assisted laser desorption/ionization (MALDI) time-of-flight (TOF) has been established as the prominent MS method due to its high workflow efficiency (i.e., low turnaround time and reduced cost per identification) [2,3]. This is also partially achieved through direct deposition of microorganism colonies on the MALDI target plate, where they are overlaid by a matrix solution without requiring additional sample manipulation.

Colony picking and depositing have traditionally been done manually with loops, toothpicks, or pipette tips [4,5]. However, since mistakes may not be apparent until much later, experienced technicians are usually required for picking the right colony. Different automated systems have been developed to reduce the burden on these technicians [6]. Automated systems allow greater sample throughput and reduce the error rate [7,8]. Although multiple colony-picking robot designs were proposed in the past, most focused on sample deposition in microtiter wells [9,10,11,12], while only a few focused on deposition on target plate required for MALDI-TOF analysis [8]. Some automated systems are room-sized, expensive, and provide high sample throughput, which most laboratories might not require [6]. Even the tabletop designs [8], which are cheaper and smaller, have the downside of being designed only for a specific task (e.g., colony picking and deposition).

The developments in the field of collaborative robots present new opportunities for laboratory automation. Collaborative robots are designed for safe robot applications within the operator’s proximity. Their integrated force sensors detect possible collisions and enable the operator to move the robot by grabbing it and moving individual segments. Therefore, operators can teach the robot a new task by demonstrating it instead of programming it. This approach is called kinesthetic teaching and is one of the learning-from-demonstration (LfD) approaches [13,14]. Most research focuses around coarse tasks where the required accuracy of the demonstration is still small enough for users to successfully transfer the desired movement on a robot, using kinesthetic teaching [15,16,17,18]. However, in the laboratory environment, many tasks require fine movements, where sub-millimetre accuracy is required. Teleoperation [19,20,21,22] and cooperative robot tools (CRT) [23,24] are the two established approaches for the demonstration of fine movements. However, they require additional input devices to control the robot, which inexperienced operators cannot use sufficiently. Additionally, it has been shown that although the two approaches perform better, it is possible to achieve sub-millimetre accuracy with the kinesthetic teaching as well [25]. Therefore it might be possible to successfully integrate collaborative robots into the laboratory environment by teaching different tasks (e.g., colony picking) using kinesthetic teaching.

However, the demonstration must be generalized to adapt the task to different conditions (e.g., different start/end positions, via points). Many different methods tackle task generalization [26,27,28,29]. Dynamic movement primitives (DMP) is a method that generalizes the task trajectory in the form of a second-order differential equation, for which only one demonstration is needed [26]. Although it cannot adapt the task trajectory to intermediate via points, it can adapt it to any starting and end position (i.e., in the case of the colony-picking task, the trajectory end position is adjusted based on the selected bacterial colony position).

An important thing to consider when adapting the collaborative robot to different tasks is its end-effector tool. Due to considerable differences in tasks, it is necessary to have specialized robot manipulator end-effector tools that can be replaced when needed. For example, the colony-picking task requires an end-effector that allows precise colony position detection, picking, and deposition.

In our study, we analyzed whether collaborative robots could be used in the laboratory environment to execute fine tasks, demonstrated by the operator using kinesthetic teaching. As a case study, the colony picking and depositing for MALDI-TOF MS analysis was chosen. We developed a specialized end-effector tool capable of precise colony detection, picking and deposition. Precise colony position detection was enabled by a combination of an RGB camera and a high-precision 2D laser profile measurement system that allowed the construction of accurate colony 3D models. A stainless steel needle was used to pick and deposit the colonies. We used the DMP method to generalize the picking and deposition task. We picked and deposited 56 bacterial colonies and identified them using Bruker Biotyper MALDI-TOF mass spectrometer to determine the system’s performance. Based on the identification scores, we determined the system’s error rate.

## 2. Materials and Methods

### 2.1. Experimental Setup

The experimental setup consisted of robot manipulator Panda (Franka Emika, GmbH, Munich, Germany), mass spectrometer MALDI Biotyper (Bruker, Inc., Billerica, MA, USA), a thermal sterilizer SteriMax smart (WLD-TEC, GmbH, Arenshausen, Germany), and disposable MALDI target plates MBT Biotarget 96 (Bruker, Inc.). The end-effector tool attached to the robot manipulator consisted of an RGB camera a2A3840-45ucPRO (Basler, GmbH, Ahrensburg, Germany), a 2D laser profile measurement system ZG2-WDS8T (Omron Industrial Automation, Inc., Hoffman Estates, IL, USA), force sensor Nano17 (ATI Industrial Automation, Inc., Apex, NV, USA), a feedback 360° high-speed servo motor (Parallax, Inc., Rocklin, CA, USA), and a stainless steel needle (Figure 1).

The Panda robot was used as a manipulator while also serving as an input device during kinesthetic teaching. Using both the RGB camera and the 2D laser profile measurement system, we detected the bacterial colony and determined its position with sub-millimetre accuracy. The RGB camera acquired an image of the Petri dish (Figure 2a), based on which the operator could determine an approximate position of the selected bacterial colony. The area around the approximate position was then scanned with the 2D laser profile measurement system, which provided the operator with a detailed 3D model of the area (Figure 2b), including the bacterial colony. The scan area dimension was 8 × 4 mm. The 2D laser profile measurement system resolution is 13 μm in *x* direction and 1 μm in *z* direction. Defined by the robot movement, the generated 3D model resolution in *y* direction was 40 μm. The generated 3D model provided the operator with the colony’s exact position. The colonies were picked and deposited using a stainless steel needle (Figure 2c). For the picked colonies to be analyzed with MALDI Biotyper mass spectrometer, they first had to be deposited on MBT Biotarget 96 target plates (Figure 2d). However, the contact forces were significant due to the target plate’s stiffness. Therefore a force sensor was used to adjust the deposition trajectory in a vertical direction based on the measured forces. For successful deposition, it was also necessary to rotate the needle. This was achieved by using a 360° servo motor with position feedback. The position feedback was necessary for an accurate picking of bacterial colonies. After each colony deposition, the needle was sterilized using 96% denatured ethyl alcohol and a thermal sterilizer SteriMax smart.

High-level control was managed using MATLAB Simulink 2019b (The MathWorks, Inc., Natick, MA, USA). It was used to connect individual units (e.g., robot manipulator, RGB camera, 2D laser profile measurement system), handle the experiment workflow, and save the measured data from Panda robot (end-effector position and orientation, joint positions, joint velocities, etc.), Nano17 force sensor (force and torque vectors), RGB camera (acquired image), and 2D laser profile measurement system (3D model). The data from the Panda robot and the force sensor was acquired with a 1 kHz sampling frequency.

A crucial issue that also had to be overcome was the robot’s kinematic model accuracy. We calibrated the robot’s Denavit-Hartenber parameters in order to reduce the average end-effector’s positioning error to 0.1 mm [25].

### 2.2. Learning from Demonstration and Trajectory Generation

Learning from demonstration is an approach where the operator demonstrates the task instead of programming it. So-called movement primitives are used to learn from the demonstrations. Dynamic movement primitives (DMP) is one of many different methods that can be used [26]. It generalizes the motion by encoding the trajectory as a second-order spring-damper system with an additional non-linear function that determines the shape of the trajectory:(1)$$\tau {\ddot{p}}_{D}\left(t\right)=\alpha_{P}\left(\beta_{P}\left(g-p_{D}\left(t\right)\right)-\tau {\dot{p}}_{D}\left(t\right)\right)+f\left(x\right),$$
where τ represents the temporal scaling term, $p_{D}\left(t\right)$, ${\dot{p}}_{D}\left(t\right)$, and ${\ddot{p}}_{D}\left(t\right)$ represent the DMP’s position, velocity, and acceleration, g a point attractor of the second-order spring-damper system, and parameters $\alpha_{P}$ and $\beta_{P}$ defining the dynamics of the spring-damper system. f(x) is the non-linear function representing the forcing term and determines the shape of the trajectory. It approximates the demonstrated trajectory with a linear combination of multiple Gaussian basis functions and their corresponding weights, where the weights are taught from a demonstration using linear regression. To avoid explicit time-dependency, it is a function of a phase variable x. Phase variable monotonically decreases towards 0, at which point the forcing term diminishes. Therefore we can be sure that for any starting position, the system will generate a trajectory that is forced by a forcing term f(x), that the trajectory will be smooth, and that it will finish at g when the forcing term vanishes. DMP can encode arbitrary movement information (e.g., position, orientation). However, for our study, it was sufficient to encode only the movement position as seen in Figure 3.

There were two main reasons why we decided to use a DMP. Firstly, it requires only one demonstration. This is necessary for the collaborative robots in the laboratory environment since the tasks should be taught in as little time as possible. Secondly, it is possible to adapt the learned task to different start and end positions while generating a smooth trajectory. In our case study, we use this to adapt the end position during the picking process and the start and end position during the deposition process (Figure 3).

During the colony deposition, however, the DMP-generated trajectory was additionally compensated in order to avoid significant contact forces. Since the contact forces had to be small (e.g., bellow 1 N), we additionally compensated the trajectory in the vertical direction based on the input from the force sensor placed on the end-effector as seen in Equation (Equation 2) to achieve satisfactory deposition.
(2)$$p_{R}\left(t\right)=p_{D}\left(t\right)+K\int f\left(t\right)\;dt;\;K=\left[\begin{array}{c} 0\\ 0\\ K_{Z} \end{array}\right],$$
where $p_{R}\left(t\right)$ represents the robot position, $p_{D}\left(t\right)$ the DMP-based reference position, and f(t) representing the force input measured by the Nano17 sensor. K represents an force control gain.

Using the DMP method and the force compensation allowed us to achieve satisfactory picking and deposition of a bacterial colony.

### 2.3. Experiment Protocol

For our experiment, we picked and deposited 56 colonies of two different bacterial species (Gram-negative *Acinetobacter baumannii*, Gram-positive *Staphylococcus epidermidis*) cultured on blood agar. At the start of the experiment, the operator demonstrated the picking process, while the deposition process was generated programmatically. Both demonstrations were then generalized using a DMP (Figure 3). Then, before each picking process, the operator selected the colony to be identified from an image. After its 3D model was generated, it determined its exact position by selecting the colony’s peak. The selection process could be automated but was performed manually for the purposes of this study. The operator defined the spot on the target plate on which the extracted colony should be deposited, after which the robot manipulator picked and deposited the colonies. The end-effector needle was also rotated around its axis during the deposition task to ensure as uniform a deposition as possible. Then, the needle was sterilized using 96% denatured ethyl alcohol and a thermal sterilizer. One deposition was made for each pick. They were then manually overlaid with the matrix solution. After a short drying period, the target plate was placed in the MALDI Biotyper mass spectrometer, and identification of each target was performed. Each spot was given, together with the identification, a corresponding score, indicating the reliability of the identification.

## 3. Results

In the following chapter, we present the results, which are relevant for the discussion. We present separate results for the picking, deposition and identification process. While the identification process results will provide us with the overall performance of our system, the picking and the deposition process will give us additional insight into the performance of those crucial steps, which will help us evaluate the system additionally.

### 3.1. Picking Process

In order to evaluate the picking process, we acquired multiple colony 3D models before and after the picking process. We plotted these models using a top-down heatmap (Figure 4). While each pixel position represents a different coordinate on a 3D model’s *x*-*y* plane with the dimensions of 8 mm × 4 mm, its color represents the 3D model’s *z* value in each coordinate. The color changes from blue to yellow depending on the *z* value, where deep blue depicts the lowest *z* values, and bright yellow depicts the highest *z* values. We omitted color legends that would provide exact *z* values to increase the figure’s clarity. We believe the absolute *z* values, in this case, are not so relevant since we are more interested in the overall change in shape. However, as a reference, the colony, which was by far the biggest and much bigger than the rest (Figure 4, top–left), had maximum dimensions of 1.7 × 2.1 × 0.5 mm.

To evaluate the picking process, we also calculated the difference in the colony volume. We defined an area under which the volume should be calculated for each colony. The calculations were done on six different colonies, which were a good representation of the whole population. In Figure 4 and in Table 1, they were sorted from left to right based on the initial volume.

### 3.2. Deposition Process

The deposition quality depended heavily on the contact force between the end-effector needle and the MALDI target plate. Position compensation was implemented in the vertical axis to limit the contact force. On the left plot of Figure 5, the contact force is plotted alongside the *z* position. Both are presented with a variance plot since the data was taken from all 56 depositions. The mean force rarely exceeds 1 N, mostly during the initial contact. After the initial contact is compensated (trajectory phase greater than 0.3), the mean force is 0.29 N ± 0.35 N. The corresponding mean pose compensation is around 0.5 mm.

The final result of deposition can be observed on the right image of Figure 5. The image was taken before the identification process when the samples had already hardened, and the layer of matrix solution was put over them.

### 3.3. Identification

The picked and deposited colonies have been identified with MALDI Biotyper mass spectrometer. Each deposition got an identification of the organism and a corresponding score indicating the level of confidence in the identification. While scores above 2.0 are considered highly reliable, identification is considered invalid when scores are lower than 1.7, even if some organism is identified [8].

For our experiment, we extracted 56 bacterial colonies in total. Of these, 36 were *A. baumannii* and 20 *S. epidermidis*, and 6 were considered invalid (5 *A. baumannii* and 1 *S. epidermidis*) as their scores were too low. The average scores with the corresponding standard deviation are presented in Table 2. The scores were presented separately for successfully identified samples and all samples since the low scores of invalid samples skewed the average score and, therefore, the picture of general performance.

## 4. Discussion

In our study, we tested whether collaborative robots, together with kinesthetic teaching, are a viable option for tasks that require sub-millimetre positioning accuracy. Our case study was a colony picking and identification process where individual bacterial colonies are picked from the Petri dish, deposited on the MALDI target plate and identified with a MALDI-TOF mass spectrometer. The study, therefore, consisted of multiple independent problems. Not only was it a question of whether the task can be demonstrated and generalized accurately enough, but also whether the available sensors and end-effector design will allow for a good adaptation of tasks to different conditions. The overall success can be measured with the number of successful identifications. However, results from the picking and deposition process give us additional insight.

The picking process was analyzed on six different colonies. In order to analyze them, we used the generated 3D models of bacterial colony area before and after the picking. It can be seen that in all cases, the colony almost completely disappears (Figure 4). Only in the case of the biggest colony, its shape is still visible, although reduced in size. The initial dimensions of that colony were too big for its whole volume to be picked in one iteration. Moreover, in most cases, the area where the colony used to be is not darker than its immediate neighbouring area. That means that the picking was done accurately, without any unnecessary picking of the agar layer under the colony. All this can be additionally confirmed by the measured volumes (Table 1). In all cases, the bacterial colony volumes were significantly reduced, where only in some cases there was some extraction of the agar layer as well, but in insignificant volumes. This is also the case for the bacterial colony with the smallest volume, where its dimensions were so small that it was physically impossible not to pick some of the agar layer as well. The absolute differences in picked volumes can be explained by the structure of the bacterial colonies themselves since they are prone to stick together due to their viscosity. All in all, the picking process results show that it is possible to demonstrate a fine task movement and successfully generalize it using a DMP.

Regarding the deposition process, we can see from the plot (Figure 5, left) that the force compensation works well. Throughout the process, the *z* position is actively compensated. Smaller contact forces allowed better deposition and prevented the damaging of equipment. During the initial contact, the contact force exceeds 1 N but is consistently compensated so that the mean force after the initial contact is approximately 0.3 N. However, if we look at the image of the deposited colonies (Figure 5, right), we can see that although the samples are all within their designated spots, they are deposited differently. Most are distributed uniformly over the whole spot, while some remain concentrated in smaller areas. The concentrated areas are not desirable since they can cause poorer identification results. However, after a visual analysis, we could not determine the features that could reliably predict poor identification results. For example, samples A1 and A2 achieved excellent identification scores with their samples being uniformly distributed, while at the same time, sample A4 was not identified, although it achieved similar levels of uniformity. On the other hand, samples C3 and C4 obtained excellent scores without achieving similar levels of uniformity as A1 and A2. That means that additional factors besides the sample distribution might influence the final identification process. Since the deposition process was consistent (e.g., movement trajectory, forces), the variation in deposition quality can be partially attributed to the physical structure of colonies and their placement on the picking needle. We mitigated the second issue by modifying the shape of the deposition trajectory and rotating the needle around its axis throughout the deposition process. We modified the deposition trajectory by defining it programmatically instead of using a human demonstration. During the testing, demonstrations were used to define the deposition trajectory as well. However, since the deposition process was unreliable, we decided to use a well-defined trajectory in order to reduce the number of potential influences on the quality of deposition. This trajectory was then used also during the main experiment. Although a pre-programmed trajectory was used as a demonstration, DMP still proved to be an appropriate method to generalize a movement since it adapted the taught movement to different start and end positions.

Furthermore, the identification results could be improved by additional deposition sample treatment. In their study, Chudejova et al. significantly reduced the identification error rate with additional treatment of deposited sample by applying 1 microliter of 70% formic acid. Whether they deposited the sample first and covered it with formic acid (e.g., semi-extraction), or they first pipetted the formic acid and then suspended the colony sample in the acid (e.g., manual wet deposition), the identification error rate dropped significantly. Therefore, we could expect an additional decrease in the identification error rate by adopting either the semi-extraction or the manual wet deposition method. When using the direct spotting method (no additional formic acid), we achieved the identification results presented in Table 2, where almost 11% of the samples could not be identified (13.9% of *A. baumannii* and 5% of *S. epidermidis*). However, these results are comparable to the manual direct spotting of Chudejova et al. We conclude that our system can consistently pick and deposit bacterial colonies, with final identification results comparable to the manual deposition. However, the identification results could be additionally improved by choosing the semi-extraction or wet-deposition method.

## 5. Conclusions

We used a collaborative robot to study whether it is possible to use kinesthetic teaching for tasks where sub-millimetre accuracy is required. We chose a medical microbiology laboratory application for the case study, specifically a bacterial colony picking and identification process using mass spectrometry. We designed a specialized robot end-effector that, besides the picking and deposition, also enabled the localization of bacterial colonies, creation of their 3D model and control of contact forces. The tasks were demonstrated by an operator and then generalized using a DMP. Although the workflow of colony detection, picking and deposition worked well, the overall identification error rate was almost 11% (13.9% of *A. baumannii* and 5% of *S. epidermidis*). The error rate is comparable to the manual deposition, however, it could be additionally reduced by applying semi-extraction or wet deposition method. We conclude that we successfully demonstrated fine tasks in the microbiological laboratory environment using kinesthetic teaching and generalizing them using DMPs. However, further system improvements are possible by modifying the deposited sample treatment.

## Figures and Tables

**Figure 1 sensors-22-02862-f001:**
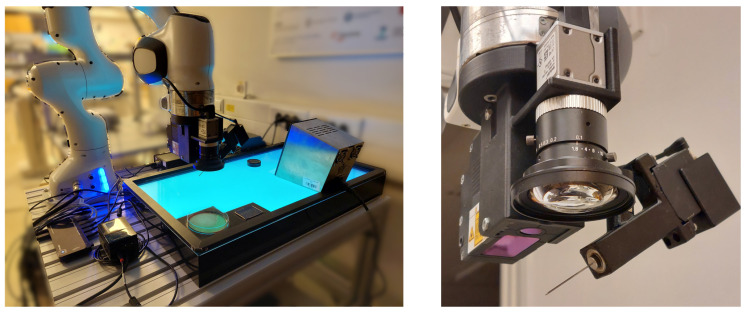
Overview of the experiment setup (**left**) and the end-effector tool (**right**).

**Figure 2 sensors-22-02862-f002:**
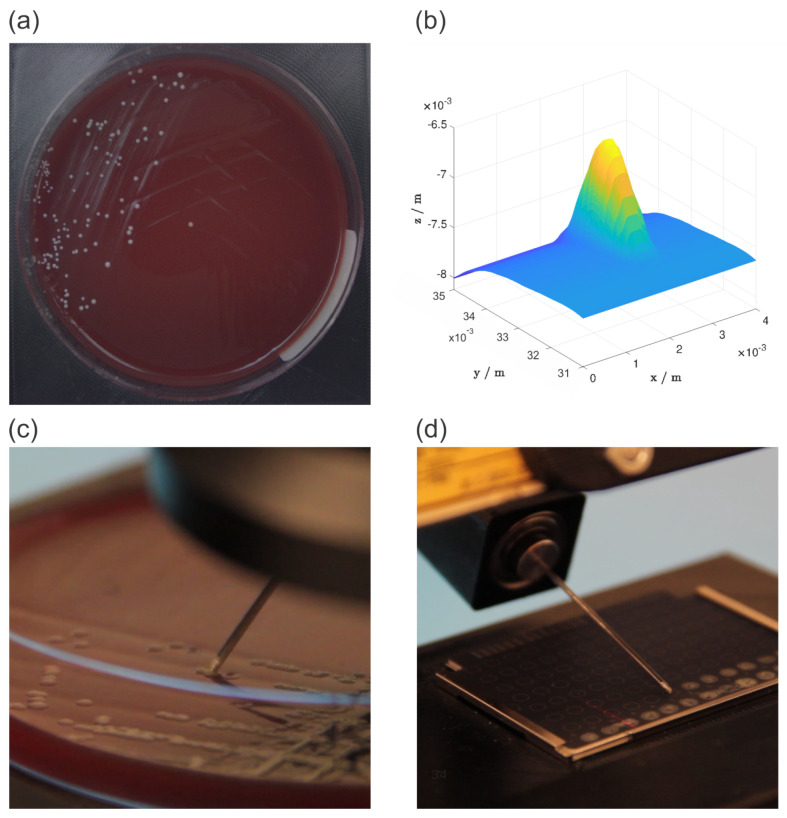
Overview of the experiment workflow: an image of a petri dish acquired with RGB camera (**a**), a 3D model of the selected bacterial colony generated from 2D laser profiles (**b**), the bacterial colony picking process (**c**), and the bacterial colony deposition process (**d**).

**Figure 3 sensors-22-02862-f003:**
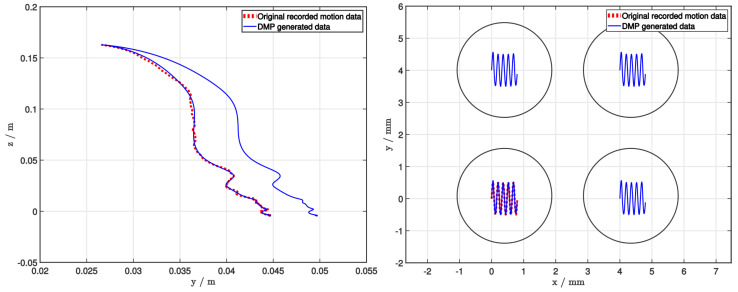
Using DMP to generalize demonstrated trajectories of the picking (**left**) and deposition (**right**) task. In order to demonstrate the quality of generalization, the end position of the picking process is modified, whereas both the start and end position are modified for the deposition process.

**Figure 4 sensors-22-02862-f004:**
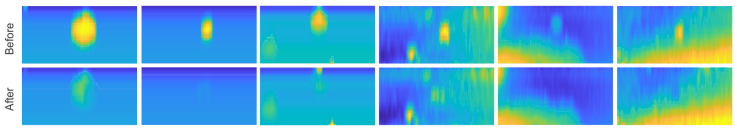
Overview of the picking process. Multiple different colonies are plotted before (**top**) and after (**bottom**) the picking process.

**Figure 5 sensors-22-02862-f005:**
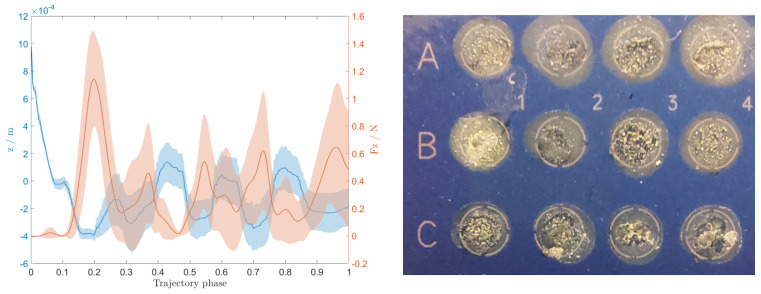
The contact force between the end–effector needle and MALDI target plate and the corresponding compensated z position (**left**), and an image of deposited colonies on the MALDI target plate after being covered with the matrix solution (**right**).

**Table 1 sensors-22-02862-t001:** Estimated colony volume change before and after the picking process.

ID	1	2	3	4	5	6
Initial volume [mm${}^{3}$]	1.44	0.78	0.46	0.11	0.06	0.02
Final volume [mm${}^{3}$]	0.41	−0.04	0.04	0.05	0.007	−0.02
Relative change [%]	−72	−105	−91	−54	−99	−193

**Table 2 sensors-22-02862-t002:** Identification process results.

	*Acinetobacter baumannii*	*Staphylococcus epidermidis*
No. of samples	36	20
No. of “score below 1.70”	3	1
No. of “no peaks found”	2	0
No. of total errors	5	1
Total errors [%]	13.9	5
Average identification score w/o errors	2.05 ± 0.21	2.06 ± 0.13
Average identification score with errors	1.94 ± 0.54	2.04 ± 0.16

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
