# Peer review of "Collaborative Robot Precision Task in Medical Microbiology Laboratory"

_sensors, 2022, doi:10.3390/s22082862_

Round 1

Reviewer 1 Report

Authors used a collaborative robot to study if it is possible to use manual teaching (or learning from demonstration) to train robotic tasks in a microbiology laboratory where precision is critical. The application is focused on a bacterial colony picking and identification process using mass spectrometry. A specialized robot end-effector has been designed and implemented.

This work is an interesting application focused on development rather than research activity. The work demonstrates that manual teaching of robots can be used to capture positions of a collaborative robot performing activities in a microbiology laboratory.

The paper does not present any novelty from a research point of view but it is interesting because it provides information regarding the accuracy that can be achieved in this application.

Dynamic movement primitives (DMP) is one of many different methods that can be used to control the task but reasons to chose it are weak other options could provide the same advantages described in the document (lines 130-135). In fact, I miss a comparative between at least two methods to know which one works better, comparing accuracy, etc.

I also miss a video showing how it works, how smooth are robot movements, etc. In this sense, any information regarding the speed processing (compared with a human) is presented. Is the robot as fast as a person doing the task?

Author Response

Dear Reviewer,

Let me first thank you for your input and comments regarding the paper since they were on point.

We could indeed use different methods. First of all, the motion could be generated as a typical point-to-point movement used in many robot applications. However, since it is important in our application to follow a specific trajectory shape, we used Dynamic movement primitives. They are used to generalize a trajectory based on a demonstration and adapt it to arbitrary start and end positions. Other methods such as ProMP or GMMs can also generate a trajectory from a demonstration. However, since they are probabilistic, they were developed to generalize based on several demonstrations as they provide an average trajectory and the corresponding deviation. Our experiment required only one trajectory, as it can not be expected from laboratory personnel to demonstrate many trajectories due to the lack of time. Therefore, we decided to use the DMP method since it was "just right" for our needs, without any unnecessary additional functionality. Moreover, since the whole workflow, in general, contained many different elements (demonstration, generalization, detection, etc.), we believe it was best to choose the methods we thought to be most relevant for each part of the experiment, implement them, test them on the whole workflow, and present the results without additional method comparison that would additionally prolong the article and dilute its message (which should be about the appropriateness of kinesthetic teaching). We hope this response provides a better insight into our decision-making when choosing the DMP method. Additionally, we believe it would be confusing to add text explaining why other methods were not chosen instead of providing additional reasons why the DMP method was chosen (there are no other reasons) and would prefer that the explanation (lines 130-135) stays as it is. However, we, of course, welcome additional feedback.

Regarding the videos, I can provide you with a picking task video so that the movement's smoothness can be observed. This is a link to my Drive folder: https://drive.google.com/file/d/1MhRwWdbTjAJI7GcEUFqSP2oP20gzhW9e/view?usp=sharing

Furthermore, let us stress that this study was made as a proof of concept and not an implementation in the real-world environment. Therefore speed was secondary and was not optimized or measured. However, a picking move where the robot manipulator moved for around half a meter was set to last for 30 seconds, whereas a deposition move was set to take 10 seconds. These times could, of course, be reduced if that would be desired.

To conclude, let us again thank you for your input, and we hope that our responses have sufficiently addressed the issues raised during your review.

Kind regards,

Aljaž Baumkircher and co-authors

Reviewer 2 Report

    The authors presented an automated solution for microbiological sample picking and depositioning with a collaborative robot arm. To achieve a more precise positioning they calibrated the robot kinematics model. Kinematic calibration of the robot arm would be more detailed. The average position error of the endefector is 0.1mm which is pretty good for collaborative arm. As described the resolution of 3D scan is 0.013mm X 0.04mmm X 0.001um where y dimension (0.04um) is caused by robot motion. How did you achieve this  resolution with 0.1 end effector position error? 3D model accuracy.  The results of picking and deposition are well structured.  A manual process measuring with the same sample would be useful to compare with system performance.    On figure 2/b the units of the axis are different, it is hard to understand.    Figure 3/b the motion data in mmx10^-3 all others mX10^-3, I think it is misspelling. 

Author Response

Dear Reviewer,

Let me first thank you for your input and comments regarding the paper since they were on point.

In general, an in-depth description of the calibration process can be accessed in our previous study as part of the supplementary files (Baumkircher et al., 2021). Regarding the scan resolution, it is true that there is some deviation from the 0.04 mm resolution value. However, this deviation should not be confused with the 0.1 mm end-effector position error. During the scanning process we moved the robot for 4 mm with constant velocity during which we acquired 100 scans. We can expect the final position to offset for 0.1 mm, but that is about it. Therefore, the resolution is defined as (4 mm +- 0.1 mm) / 100, which is approximately 0.04 mm.

A manual process measuring with the same sample would be useful, but we decided against it since other research papers thoroughly analysed the manual deposition. Because those researchers used similar bacteria, we decided to use all available samples to test the workflow presented in the article. We still used manual workflow to learn and improve some segments of the automated workflow (e.g. deposition process).

Regarding the figures, we agree that figure 2b can be misleading since some units are x10-3 and some are not. The reason for that is that this specific plot represents an actual colony that was scanned somewhere in space. And the issue is that when formatting the text, MATLAB prefers to write 0.035 instead of 35x10-3. We modified it so that it is consistent. Regarding figure 3b, it was a misspelling, and we removed it. Thank you!

To conclude, let us again thank you for your input, and we hope that our responses have sufficiently addressed the issues raised during your review.

Kind regards,

Aljaž Baumkircher and co-authors

Reviewer 3 Report

The paper treats the method of implementation of collaborative robots in a medical microbiology laboratory for bacterial colony picking and identification.

I have the following remarks:

Theoretical support for high-precision positioning techniques needs to be developed.

A detailed mechanical description of the specialized end-effector tool is required

The figures are unclear (Fig 2) and insufficiently explained

The paper treats the method of implementation of collaborative robots in a medical microbiology laboratory for bacterial colony picking and identification.

Author Response

Dear Reviewer,

Let me first thank you for your input and comments regarding the paper.

However, we are a bit confused by the first remark since we do not understand for what exactly the theoretical support should be developed. To achieve high precision, we successfully calibrated robot’s Denavit-Hartenberg parameters (Baumkircher et al., 2021, supplementary files). Additionally, the other relevant topic is the positioning precision during the operator’s demonstration. This article focuses on whether kinesthetic teaching is applicable in an application where precise movements are required (e.g. colony picking). The main issue with kinesthetic teaching is the imperfect compensation of movement dynamics, making it harder for users to demonstrate precise tasks. The imperfect compensation is inherent to any robot manipulator and differs between different robot manipulator series and manufacturers. That is also one of the main reasons both CRT and teleoperation approaches slightly outperform kinesthetic teaching. However, it should not be forgotten that kinesthetic teaching is relevant for this application since no additional input devices are necessary. As there was no high-precision positioning technique required for the demonstration, we believe there is no reason to develop any theoretical support. As we are not quite sure about the meaning of this remark, we apologize if we misunderstood and assumed wrongly.

Regarding the detailed mechanical description of the specialized end-effector tool, we prepared one, and it is accessible on the following link. We hope this is what you requested. If desired, we can include it in the article. However, we believe that multiple projections necessary to show all dimensions properly would clutter the article too much while providing too little relevant information and would therefore prefer not to include it. Of course, we welcome additional feedback regarding that since we might have a different opinion.

Link: https://drive.google.com/file/d/1BL_f_asrDmg_4ibZlW0haefLRjPEamw8/view?usp=sharing

Regarding the figures, they are described in detail in lines 87-98. However, we modified the caption so that the description is more comprehensive.

To conclude, let us again thank you for your input, and we hope that our responses have sufficiently addressed the issues raised during your review.

Kind regards,

Aljaž Baumkircher and co-authors